# Transparent Machine Learning Reveals Diagnostic Glycan Biomarkers in Subarachnoid Hemorrhage and Vasospasm

**DOI:** 10.3390/ijms26167727

**Published:** 2025-08-10

**Authors:** Attila Garami, Máté Czabajszki, Béla Viskolcz, Csaba Oláh, Csaba Váradi

**Affiliations:** 1Institute of Energy, Ceramic and Polymer Technology, University of Miskolc, 3515 Miskolc, Hungary; attila.garami@uni-miskolc.hu; 2Department of Neurosurgery, Borsod-Abaúj-Zemplén County Center Hospital and University Teaching Hospital, 3526 Miskolc, Hungary; czamate@gmail.com (M.C.); olahcs@gmail.com (C.O.); 3Institute of Chemistry, Faculty of Materials Science and Engineering, University of Miskolc, 3515 Miskolc, Hungary; bela.viskolcz@uni-miskolc.hu

**Keywords:** serum glycosylation, N-glycans, mass spectrometry, liquid chromatography, subarachnoid hemorrhage, cerebral vasospasm, decision tree, interpretable machine learning

## Abstract

Subarachnoid hemorrhage (SAH) and its major complication, cerebral vasospasm (CVS), present significant challenges for early diagnosis and risk stratification. In this study, we developed interpretable decision tree models to differentiate between healthy controls, SAH patients, and SAH patients with vasospasm using serum N-glycomic data. Building on previously published glycomic profiles, we introduced a refined modeling approach combining systematic preprocessing, feature selection, and interpretable machine learning. Our methodology included outlier removal, standard scaling, and a novel correlation-based feature reduction guided by feature importance scores derived from preliminary decision trees. Binary classification tasks (Control vs. SAH and Control vs. CVS, and SAH vs. CVS) were evaluated through stratified repeated cross-validation and hyperparameter optimization. Models achieved high accuracy (up to 0.91) and stable F1-scores across configurations. Key glycans such as FA2(6)G1 (bi-antennary, fucosylated, monogalactosylated), A4G4S3(2) (tetra-antennary, tetra-galactosylated, tri-sialylated), and A3G3S3(5) (tri-antennary, tri-galactosylated, tri-sialylated) emerged as the most discriminative. Visualizations that combine joint feature distributions and decision boundaries provided intuitive insight into the classifier’s logic. These findings support the integration of interpretable glycomics-based models into clinical workflows.

## 1. Introduction

Subarachnoid hemorrhage (SAH), primarily caused by the rupture of intracranial aneurysms, is a critical condition characterized by high mortality rates and prolonged neurological disabilities [1]. A significant complication arising from SAH is cerebral vasospasm (CVS), which greatly contributes to delayed cerebral ischemia and adverse patient outcomes [2]. Therefore, early and accurate identification of patients at risk for vasospasm is crucial, especially during the initial phase post-hemorrhage, when traditional imaging or neurological evaluations often yield inconclusive results [3].

Recent research has shown that glycosylation patterns, particularly N-linked glycans on serum glycoproteins, are altered in various cerebrovascular disorders, including ischemic stroke and peripheral vascular diseases [4]. Profiling serum N-glycome presented a promising avenue for biomarker discovery, capturing subtle pathophysiological changes linked to inflammatory and vascular processes [5,6]. Building on our previously published findings regarding changes in serum N-glycosylation patterns in SAH and CVS, we have expanded our data analysis approach using interpretable machine learning techniques [7] The study identified significant differences in the glycosylation patterns between the patient groups and healthy controls. Notably, higher levels of sialylation and specific glycan structures, such as A2G2 (biantennary-bigalactosylated), FA2G2 (biantennary-bigalactosylated fucosylated), A2G2S1 (biantennary-bigalactosylated, monosialylated), and A3G3S2 (tri-antennary tri-galactosylated bi-sialylated), were associated with SAH and CVS. These altered glycan profiles suggest the potential utility of glycosylation analysis as a diagnostic tool for identifying patients at risk of developing CVS after SAH. The findings support the notion that glycan alterations are reflective of underlying pathophysiological changes in these conditions.

Machine learning (ML) methods have garnered increasing interest in the realm of clinical biomarker research due to their capability to uncover complex, non-linear relationships within high-dimensional datasets [8]. However, many ML models lack interpretability, posing challenges for clinical adoption [9,10,11]. In contrast, decision trees provide a balance between predictive accuracy and explainability, facilitating transparent decision-making and the integration of ML tools into clinical workflows [12,13].

This study evaluated the utility of serum N-glycan profiles in distinguishing healthy individuals, SAH patients without vasospasm, and those who develop vasospasm. Beyond identifying disease-specific glycan patterns relative to controls, we also implemented a direct SAH vs. CVS classification to uncover glycomic markers predictive of vasospasm within the SAH population—addressing a key gap in early diagnostics.

We utilized a straightforward, interpretable decision tree classification model, prioritizing both transparency and clinical relevance. We also examined the impact of data preprocessing—such as outlier removal, scaling, and correlation-based feature reduction informed by decision-tree feature importance—and identified key glycan structures associated with various disease states. Furthermore, we employed decision boundary visualizations and robust cross-validation to enhance interpretability and reproducibility.

## 2. Results

### 2.1. Correlation-Based Feature Filtering

As the first step of the analysis pipeline, correlation-based feature filtering was applied to reduce the number of variables and improve model interpretability. Glycan pairs with strong Spearman correlation (ρ > 0.9) were identified, and within each pair, the feature with higher importance—estimated using a preliminary decision tree—was retained.

This procedure allowed for the removal of redundant features without compromising biological interpretability or classification performance. Reducing the number of variables also simplified the structure of the resulting decision trees and improved the stability of selected glycans across cross-validation folds.

Table 1 summarizes the removed glycan pairs, their correlation coefficients, and the retained variables based on model-informed importance.

### 2.2. Classification Performance

Classification models were trained under all combinations of preprocessing: raw input, outlier removal only, scaling only, and both. Decision tree classifiers were applied to three binary classification tasks: Control vs. SAH, Control vs. CVS, and SAH vs. CVS.

The results of the Control vs. SAH and Control vs. CVS tasks are presented in Table 2. The highest performance was achieved in the Control vs. CVS comparison, with an accuracy and F1-score of 0.91 on raw data. In the Control vs. SAH task, slightly lower metrics were observed, with a maximum accuracy of 0.81. Preprocessing did not improve performance in these tasks, and in several cases, a slight decrease was noted.

The SAH vs. CVS classification results are presented in Table 3. In this comparison, moderate model performance was observed, with a top F1-score of 0.78. Notably, in contrast to the previous tasks, preprocessing steps—especially outlier removal, either alone or combined with scaling—improved classification results. Scaling alone did not yield significant improvements, which aligns with the known scale-invariance of decision tree algorithms. Nevertheless, it was included as a control to confirm the lack of distorting effects. These results indicate that subtle differences between the two disease groups become more detectable with appropriate data cleaning. Furthermore, the preprocessing configuration influenced the set of top-ranked features, suggesting that the model’s sensitivity to data structure extends to feature importance as well.

The rows highlighted in bold in Table 2 and Table 3 indicate the best-performing models in each binary classification task. These selected configurations were used to generate Figure 1 and served as the basis for all subsequent analyses and interpretations.

Overall, the results demonstrate that raw N-glycan profiles are sufficiently informative, particularly in distinguishing control from CVS cases. Decision tree models proved well suited to these tasks, but the performance differences and preprocessing sensitivity justify a detailed comparison.

When comparing the best configurations across the three tasks, the CVS model consistently outperformed the others in all metrics (accuracy and F1: 0.91), while the SAH and SAH vs. CVS models showed somewhat lower results (F1: 0.81 and 0.78, respectively). Figure 1 visualizes the performance of the best models in all three classification tasks across four metrics (accuracy, precision, recall, F1), showing the average performance and standard deviation across cross-validation folds, which reflect the reliability of each configuration. The CVS model not only achieved higher scores but also demonstrated greater stability.

### 2.3. Feature Importance Profiles

The most informative glycan structures were identified by computing feature importance scores based on decision tree models for each binary classification task. These scores were averaged across repeated cross-validation folds, and standard deviations were calculated to reflect the stability of feature selection. Figure 2 illustrates the average importance and variability of the top features across the three models, highlighting both shared and task-specific glycans. For comparability across models, feature importance values were normalized so that their sums equaled one, allowing consistent interpretation despite differences in model complexity or class balance.

In the CVS model, the glycan A4G4S3(2) was assigned the highest importance with minimal variability, indicating a consistently strong predictive role. In the SAH model, FA2(6)G1 was identified as the most informative, although its higher standard deviation suggested less stable behavior. In the SAH vs. CVS comparison, A3G3S3(5) and A2G2 ranked among the top features, implying their relevance in capturing finer distinctions between the two disease states. A4G4S3(2) appeared prominently across all models, indicating its potential as a robust shared biomarker.

### 2.4. Visual Interpretability

In order to enhance the interpretability of the decision tree models, key glycan features were visualized using two-dimensional jointplots and full decision tree diagrams. These visual tools illustrate how selected glycan concentrations influenced the classification outcomes, and how simple decision boundaries—derived from the trained trees—enabled discrimination between groups. The glycan pairs shown correspond to the top two most important features identified in the best-performing models for each binary task, as reported in Table 2 and Table 3.

Figure 3 displays the joint distribution of the two most discriminative glycans in the SAH classification task—FA2(6)G1 and A4G4S3(2). Clear group-wise separation is observed: SAH patients tend to have lower FA2(6)G1 and higher A4G4S3(2) levels compared to healthy controls. The decision regions, derived directly from the tree structure, reveal a simple threshold-based rule involving both features, resulting in clearly separated prediction zones. Marginal histograms show the distribution of each feature individually by class.

This compact classification logic—using only two glycan thresholds—underscores the explanatory power of the model and supports the potential diagnostic relevance of FA2(6)G1 and A4G4S3(2).

Figure 4 shows the corresponding decision tree trained on the full dataset for SAH classification. The first split is made on FA2(6)G1 (≤2.71%), effectively isolating a group of SAH patients. A secondary split on A4G4S3(2) (≤0.625%) completes the classification, resulting in terminal nodes with high purity. The shallow depth and small number of decision rules highlight the interpretability and diagnostic potential of these features.

Figure 5 displays the jointplot for the CVS classification task, using A4G4S3(2) and A2G2—the top features selected by the best-performing model. The decision boundary, again extracted directly from the trained decision tree, illustrates a clear separation between control and vasospasm groups, primarily along the A4G4S3(2) axis. As seen in the marginal histograms, A4G4S3(2) levels are consistently higher in the CVS group, while A2G2 contributes more subtly to refinement of the decision.

Figure 6 shows the corresponding decision tree for CVS classification. The root split occurs at A4G4S3(2) ≤ 0.59%, effectively partitioning the samples by disease state. Subsequent splits include M5 and A2G2, both of which support clinically relevant separation. As with the SAH model, the tree remains shallow and interpretable, emphasizing the explanatory clarity provided by the selected glycans.

Figure 7 displays the jointplot for the SAH vs. CVS classification task, based on the top two features from the best-performing model: A2G2 and A4G4S3(2). The decision boundary, derived from the decision tree, separates the two groups along these axes, with marginal histograms indicating class-specific distribution shifts. Although the separation is less distinct than in previous comparisons, consistent group-level differences are apparent, and clinically relevant separation is still achieved.

Figure 8 shows the corresponding decision tree for the SAH vs. CVS task. The initial split occurs at A2G2 ≤ 1.0%, followed by a secondary division on A4G4S3(2) ≤ 0.64%. A further split on M5 refines the classification. While the decision structure is slightly more complex, the tree remains relatively shallow, and the selected glycans demonstrate biological plausibility and stable predictive value.

Together, these visualizations confirm that the decision tree models relied on biologically plausible and interpretable glycan features, offering transparent and clinically relevant classification logic.

## 3. Discussion

This study demonstrated that interpretable decision tree classifiers can accurately differentiate between subarachnoid hemorrhage (SAH), cerebral vasospasm (CVS), and healthy controls based on serum N-glycan profiles. The models achieved high predictive performance, particularly in the CVS vs. Control comparison, with consistently robust F1-scores and minimal variability across cross-validation folds. This superior performance suggests that glycomic alterations are more pronounced in vasospasm than in SAH alone, likely reflecting distinct inflammatory or vascular remodeling mechanisms. Notably, the CVS vs. Control model achieved an average F1-score of 0.91 with a standard deviation of 0.08, while the SAH vs. CVS task reached a lower average of 0.78 ± 0.16, illustrating the greater classification challenge in differentiating the two pathological conditions. In contrast, classification between SAH and CVS yielded lower accuracy and F1-scores, indicating greater overlap in glycan patterns between these conditions, despite their clinical divergence.

The use of decision tree models offered several key advantages. Most notably, their inherent interpretability enabled the derivation of simple, threshold-based decision rules. The majority of classification tasks could be resolved with two to three glycan-based splits, facilitating transparent logic that could be easily reviewed or implemented in clinical settings. Additionally, the application of correlation-informed feature filtering supported model parsimony without compromising performance, helping to reduce redundancy while preserving biologically meaningful signals.

Among the most informative glycans, A4G4S3(2) consistently ranked as a dominant feature across all tasks, confirming its central role in distinguishing disease states. FA2(6)G1 showed strong relevance in SAH-related models, whereas A2G2 and A3G3S3(5) contributed to CVS and SAH vs. CVS classification, respectively. Interestingly, the stability of feature importance scores across cross-validation folds varied by task: A4G4S3(2) was not only highly ranked but also stable, whereas FA2(6)G1 displayed more variability, which may reflect underlying heterogeneity in SAH pathology.

Preprocessing steps—including outlier filtering and scaling—were systematically evaluated. However, these techniques did not improve classification accuracy in most cases. The exception was observed in the SAH vs. CVS task, where mild improvement was noted after outlier removal. These results are consistent with the known scale-invariance of decision tree models and reinforce the idea that decision trees are particularly robust to unscaled or minimally processed data when glycan features are well curated.

The compact and interpretable nature of the models positions them as promising tools for clinical implementation. With minimal reliance on black-box assumptions and limited feature requirements, these classifiers are ideal candidates for early-stage biomarker-driven screening in neurovascular care. The results further emphasize the potential of serum glycosylation analysis in capturing vascular and inflammatory pathophysiology, supporting the continued exploration of glycomics as a diagnostic modality in cerebrovascular disease. One key limitation of this study is the relatively small sample size (*n* = 22 per group), which may limit the generalizability of the findings. Nevertheless, the models showed stable performance across cross-validation folds, supporting the robustness of the results and motivating future validation in larger, independent cohorts.

## 4. Materials and Methods

This study employed a structured pipeline to integrate high-resolution glycomics and decision tree-based classification for the differentiation of healthy controls, SAH patients, and patients with cerebral vasospasm. The methodological design emphasized transparency and reproducibility, aligning each analytical step with clinical interpretability. Although decision tree classifiers support multi-class problems, we intentionally applied binary classification models to answer specific clinical questions (e.g., early SAH detection, CVS risk stratification) while improving model robustness given the small group sizes. Binary classification also yielded simpler and more interpretable decision rules [14]. Key elements included enzymatic glycan release and fluorescent labeling, robust chromatographic analysis, and systematic data preprocessing strategies such as scaling, outlier filtering, and correlation-based feature pruning guided by model-derived feature importance. To ensure reliable evaluation, stratified train–test splitting and repeated cross-validation were implemented alongside comprehensive performance metrics.

An overview of the full analytical workflow is shown in Figure 9, summarizing the major steps from patient grouping and sample preparation through glycan analysis, data preprocessing, model training, and interpretability evaluation. This schematic provides context for the detailed methodological steps described in the subsequent sections.


**Patient Cohorts and Sample Preparation**


Serum samples were collected from three groups: healthy controls (HC, *n* = 22), patients with subarachnoid hemorrhage without vasospasm (SAH, *n* = 22), and SAH patients with confirmed vasospasm (CVS, *n* = 22). Diagnosis of CVS was based on transcranial Doppler ultrasound (TCD) criteria. The study was conducted according to the guidelines of the Declaration of Helsinki and approved by the Institutional Science and Research Ethics Committee of B-A-Z County Central Hospital and University Teaching Hospital (protocol code IG-102-102/2018 and date of approval: 4 April 2018) based on the 23/2002. (V. 9.) EüM Decree on medical research conducted on humans.

N-glycans were enzymatically released using PNGase F (New England Biolabs, Ipswich, MA, USA), followed by fluorescent labeling with procainamide (Sigma-Aldrich St. Louis, MO, USA). Glycan purification was performed using solid-phase extraction cartridges (GL Sciences Inc., Tokyo, Japan). Chromatographic separation and detection were conducted via HILIC-UPLC-FLR-MS (Waters, Milford, MA, USA), a method widely used in glycomics for its robustness and high sensitivity to glycan heterogeneity. This analytical method was selected for its high sensitivity, resolution, and ability to detect subtle changes in glycan structure, which are crucial for identifying disease-related alterations. Detailed procedures were previously described in our earlier publication [7]. Glycan nomenclature was used as Harvey et al. [15].


**Data Acquisition and Preprocessing**


Raw peak area data for identified glycan structures were exported and processed in Python (v3.10) using pandas, NumPy, and scikit-learn. All models were built on glycan intensity features, excluding metadata to ensure purely glycomic-based predictions.

For each classification task (Control vs. SAH, Control vs. CVS, SAH vs. CVS), four pipeline variants were explored:No preprocessingScaling only (StandardScaler)Outlier removal only (IQR-based filter)Scaling + outlier removal

These combinations were included to test the robustness of the model under different data preparation assumptions.

Outliers were removed using the interquartile range (IQR) method with a threshold of 1.5. This method was selected due to its simplicity, interpretability, and wide use in biomedical signal filtering, helping to mitigate the influence of extreme values without assuming data normality.

Correlated features (Spearman rho > 0.9) were filtered using a novel approach that retained the more informative variable based on feature importance scores estimated from a preliminary decision tree classifier. This strategy was chosen because it better aligns feature selection with the model’s decision logic, enhances interpretability, and preserves features that contribute meaningfully to classification rather than relying solely on statistical dispersion.

Feature scaling was applied using scikit-learn’s StandardScaler, transforming variables to have zero mean and unit variance. Although decision trees are generally insensitive to monotonic transformations, scaling was included to explore indirect effects on feature correlation structure and to maintain compatibility with potential future comparisons using other model families. These preprocessing options were applied in all combinations (with/without outlier removal; with/without scaling) for each classification task to assess robustness and evaluate their influence on model structure, feature selection stability, and predictive performance.


**Feature Selection and Model Training**


Model performance was evaluated using 5×10 repeated stratified cross-validation, which ensured robustness while maintaining class balance across folds. This approach reduced variance caused by random splits, ensured that all samples contributed to both training and testing phases across iterations, and provided a more stable estimate of model performance by averaging results over multiple fold repetitions, which is particularly beneficial in studies with limited sample sizes.

Hyperparameter optimization was performed using grid search with a 5 × 10 repeated stratified cross-validation. Repetition was included to reduce variance in model evaluation due to random splits. Stratification maintained equal class representation in each fold, which is particularly important for biomedical datasets with limited but balanced sample sizes.

The parameter grid included:Criterion: [“gini”, “entropy”]Max depth: [3, 5, 10, None]Min samples split: [2, 5, 10]Min samples leaf: [1, 2, 5]

The best-performing model configurations were retrained on the entire dataset to support interpretability analysis and visualization. This retraining allowed the construction of final decision trees using all available data, thereby maximizing statistical power and enabling clear threshold-based rule sets.


**Performance Evaluation**


Model performance was assessed using accuracy, precision_macro, recall_macro, and f1_macro metrics. These metrics were chosen to provide a balanced evaluation of classification performance, particularly in a clinical context where both sensitivity (recall) and reliability (precision) are essential. Macro-averaging was used to ensure equal weight was given to each class, regardless of sample size, which is particularly relevant in datasets with modest but balanced group sizes, as in this study. F1-score was emphasized as the harmonic mean of precision and recall, capturing the trade-off between false positives and false negatives, which is critical in diagnostic decision-making.


**Interpretability and Feature Importance**


Feature importance analysis was conducted to determine which glycan structures contributed most to classification. Scores were extracted from decision tree models trained on each cross-validation split using the feature_importances_ attribute provided by scikit-learn. For each binary classification task, importance values were averaged across all folds, and standard deviations were computed to assess the stability of feature selection. This strategy enabled the identification of glycan features that consistently influenced decision boundaries, independent of train/test split or preprocessing configuration [2,4].

The resulting scores were visualized as bar plots, with error bars representing fold-wise standard deviations. Shared and task-specific glycans were examined to support biological interpretability, particularly in the light of previous evidence linking glycosylation to vascular and inflammatory processes [7,8,14]. Additionally, jointplots of top-ranked feature pairs were used to illustrate class separation and overlay decision boundaries. Final decision trees were retrained on the full dataset using the best-performing parameter settings and visualized to support transparent review of the model structure.

## 5. Conclusions

Serum N-glycan profiles proved effective in distinguishing patients with subarachnoid hemorrhage and cerebral vasospasm from healthy individuals and from each other. Interpretable decision tree models offered high diagnostic performance and transparent rule-based logic, relying on a small set of biologically meaningful glycan markers. The most consistently informative features—particularly A4G4S3(2), FA2(6)G1, and A2G2—demonstrated task-specific and shared relevance. Preprocessing steps such as scaling and outlier removal did not enhance model performance, affirming the strength of the raw glycomic signal. Transparent, threshold-based classification provided a foundation for integration in clinical workflows, with strong internal validity demonstrated across cross-validation folds.

These results support the broader use of glycan-based machine learning models in biomarker-driven diagnostics and risk assessment for neurovascular conditions. Future efforts should focus on external validation using independent patient cohorts and on integrating glycomic data with clinical scores, laboratory markers, and imaging features. Moreover, incorporating advanced yet interpretable machine learning techniques—such as ensemble models or explainable deep learning frameworks—may further improve performance while preserving model transparency.

## Figures and Tables

**Figure 1 ijms-26-07727-f001:**
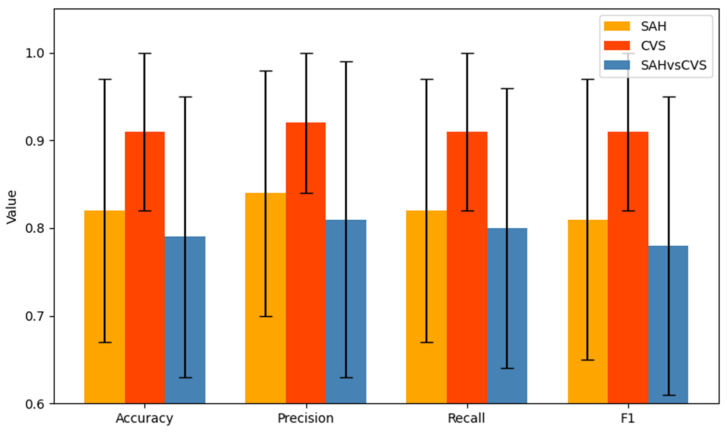
Cross-validated performance of the best decision tree models for Control vs. SAH, Control vs. CVS, and SAH vs. CVS classifications. (Mean and standard deviation of accuracy, precision, recall, and F1-score across repeated cross-validation folds).

**Figure 2 ijms-26-07727-f002:**
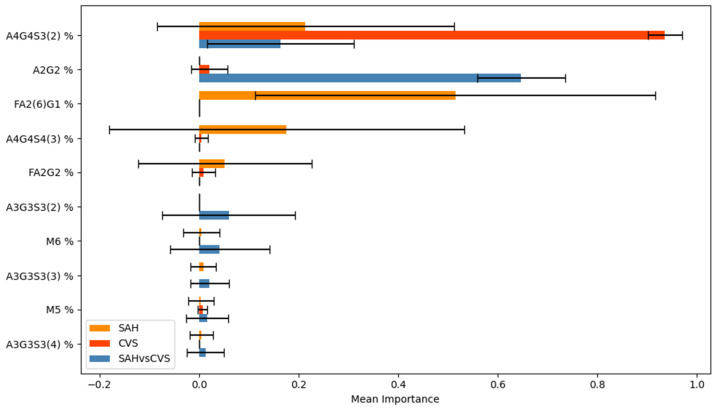
Top 10 glycan features by mean importance and standard deviation across cross-validation folds, based on the best-performing models for Control vs. SAH, Control vs. CVS, and SAH vs. CVS classifications.

**Figure 3 ijms-26-07727-f003:**
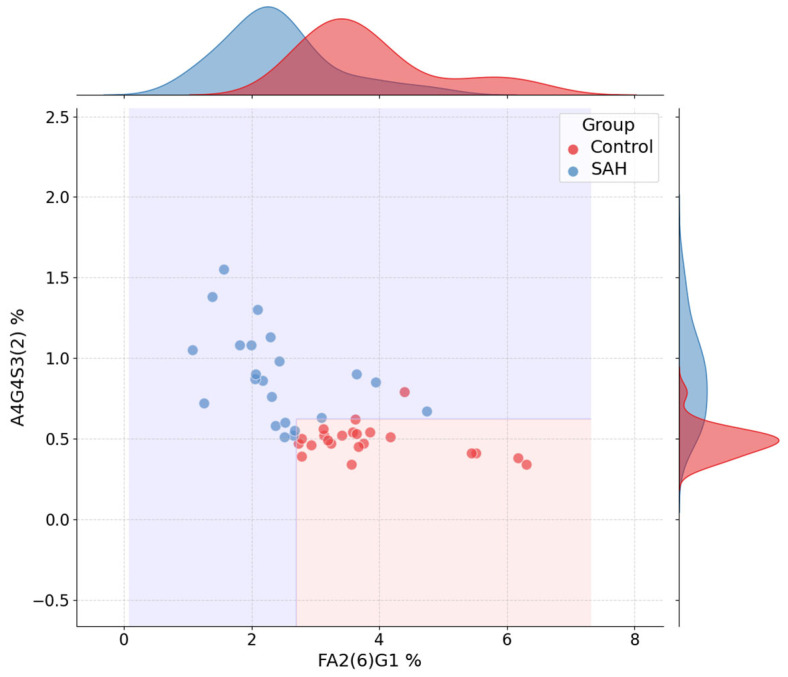
Jointplot for SAH vs. Control classification using FA2(6)G1 and A4G4S3(2). Marginal histograms and decision boundaries are shown.

**Figure 4 ijms-26-07727-f004:**
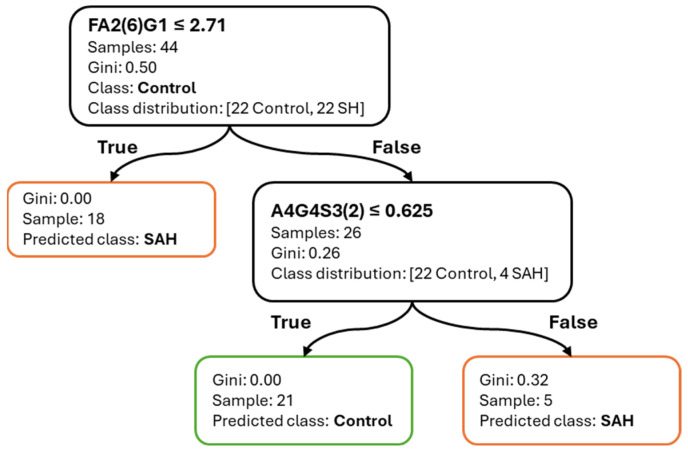
Decision tree for SAH vs. Control classification. Feature thresholds and class distributions are illustrated.

**Figure 5 ijms-26-07727-f005:**
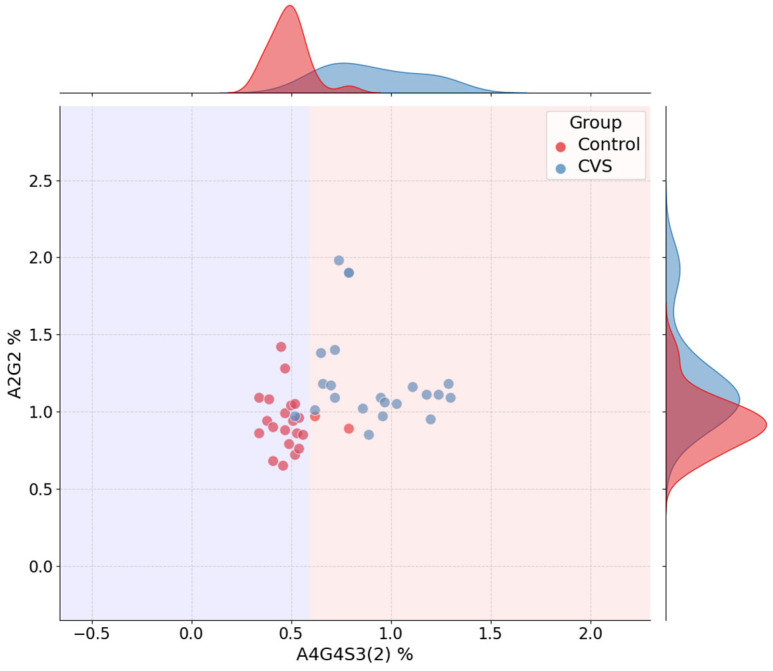
Jointplot for CVS vs. Control classification using A4G4S3(2) and A2G2. Marginal histograms and decision boundaries are shown.

**Figure 6 ijms-26-07727-f006:**
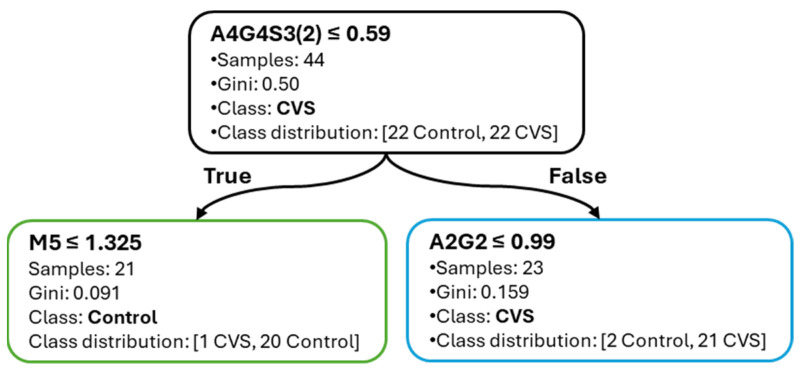
Decision tree for CVS vs. Control classification. Glycan-based splits and classification structure are visualized.

**Figure 7 ijms-26-07727-f007:**
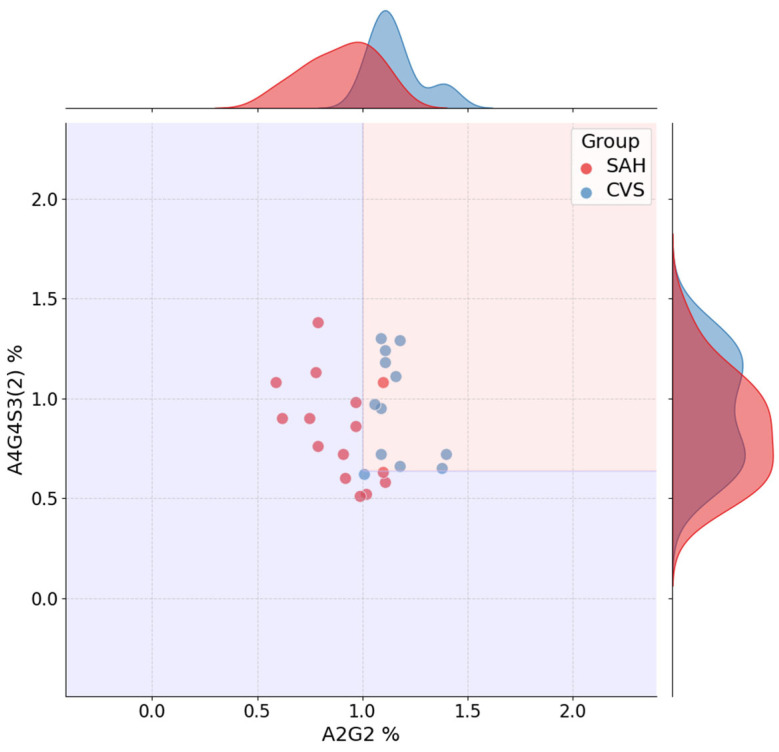
Jointplot for SAH vs. CVS classification using A2G2 and A4G4S3(2). Marginal histograms and decision boundaries are shown.

**Figure 8 ijms-26-07727-f008:**
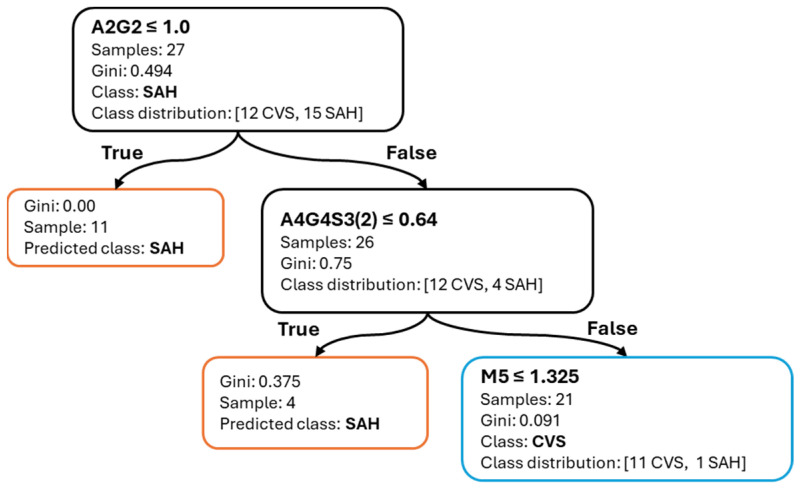
Decision tree for SAH vs. CVS classification. A2G2, A4G4S3(2), and M5 define the interpretable decision logic.

**Figure 9 ijms-26-07727-f009:**
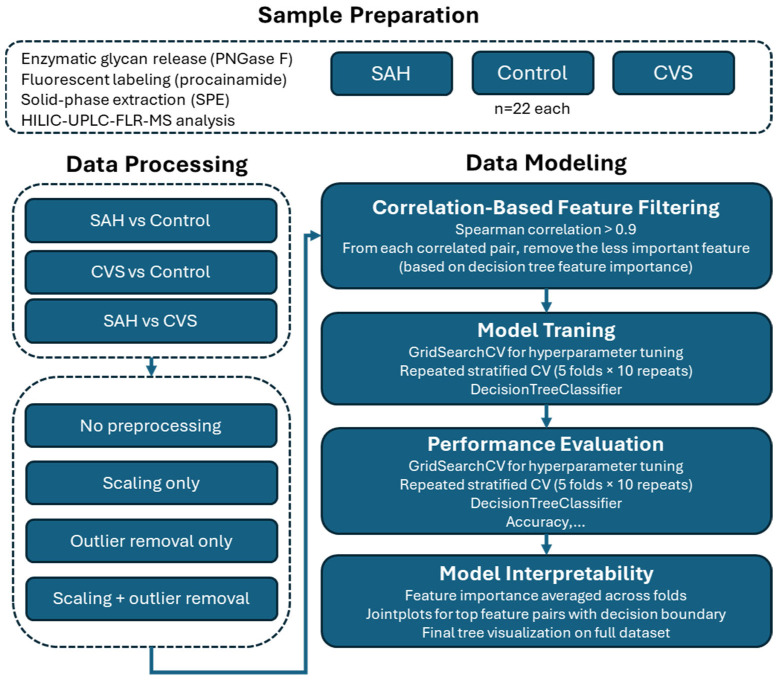
Overview of the decision tree-based classification pipeline.

**Table 1 ijms-26-07727-t001:** List of highly correlated glycan feature pairs and retained variables based on model-informed filtering [5]. FA2B: bi-antennary fucosylated with bisecting N-acetyl-glucosamine, FA2(3)G1: bi-antennary fucosylated monogalactosylated, A2G2S1: biantennary-bigalactosylated, monosialylated, A3G3S3: tri-antennary tri-galactosylated tri-sialylated, FA3G3S3: tri-antennary tri-galactosylated tri-sialylated fucosylated, A4G4S4: tetra-antennary tetra-galactosylated tetra-sialylated.

Feature 1	Feature 2	Corr	Kept	Dropped
**FA2B**	FA2	0.93	FA2B	FA2
**FA2(3)G1**	FA2(6)G1	0.92	FA2(3)G1	FA2(6)G1
**A2G2S1**	A2G2	0.94	A2G2S1	A2G2
**A3G3S3(1)**	A2BG3S2(1)	0.91	A3G3S3(1)	A2BG3S2(1)
**A3G3S3(3)**	FA3G3S2	0.96	A3G3S3(3)	FA3G3S2
**A3G3S3(4)**	A3G3S3(1)	0.91	A3G3S3(4)	A3G3S3(1)
**FA3G3S3**	FA3G3S2	0.91	FA3G3S3	FA3G3S2
**FA3G3S3**	A3G3S3(3)	0.92	FA3G3S3	A3G3S3(3)
**A4G4S4(2)**	A4G4S3(1)	0.90	A4G4S4(2)	A4G4S3(1)

**Table 2 ijms-26-07727-t002:** Classification performance and top-ranked features across preprocessing settings for Control vs. SAH and Control vs. CVS tasks. (Includes accuracy, precision, recall, F1-score, and the top two features for each configuration. Best-performing models are highlighted in bold).

Case	Out.	Scal.	Accuracy	Precision	Recall	F1	Features
**SAH**			**0** **.** **82**	**0** **.** **84**	**0.82**	**0.81**	**FA2(6)G1 %, A4G4S3(2) %**
SAH		x	0.82	0.84	0.82	0.81	FA2(6)G1 %, A4G4S3(2) %
SAH	x		0.71	0.72	0.71	0.68	A4G4S4(3) %, A2G2S2(2) %
SAH	x	x	0.70	0.71	0.70	0.67	A4G4S4(3) %, A2G2S2(2) %
**CVS**			**0** **.** **91**	**0** **.** **92**	**0.91**	**0.91**	**A4G4S3(2) %, A2G2 %**
CVS		x	0.91	0.92	0.91	0.91	A4G4S3(2) %, A2G2 %
CVS	**x**		0.86	0.88	0.87	0.85	A4G4S3(2) %, A3G3S3(5) %
CVS	x	x	0.86	0.88	0.87	0.85	A4G4S3(2) %, A3G3S3(5) %

**Table 3 ijms-26-07727-t003:** Classification performance and top-ranked features across preprocessing settings for SAH vs. CVS task.

Out.	Scal.	Accuracy	Precision	Recall	F1	Features
		0.65	0.67	0.65	0.64	FA2(6)G1 %, A2G2 %
	x	0.65	0.67	0.65	0.64	FA2(6)G1 %, A2G2 %
**x**		**0.79**	**0.81**	**0.80**	**0.78**	**A2G2 %, A4G4S3(2) %**
**x**	x	0.79	0.81	0.80	0.78	A2G2 %, A4G4S3(2) %

## Data Availability

The datasets generated and analyzed during the current study are not publicly available due to patient confidentiality and ethical restrictions but are available from the corresponding author on reasonable request and with appropriate institutional approval.

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
