# Peer review of "Transparent Machine Learning Reveals Diagnostic Glycan Biomarkers in Subarachnoid Hemorrhage and Vasospasm"

_ijms, 2025, doi:10.3390/ijms26167727_

Round 1

Reviewer 1 Report

Comments and Suggestions for Authors

Subarachnoid haemorrhage (SAH) is a serious condition which can lead to cerebral vasospasm as a primary complication. This has been shown to result in an increased mortality rate and a severe effect on prognosis. The identification of biomarkers not only aids in the early diagnosis of the disease but also contributes to the study of its pathological mechanisms. In this manuscript, the authors develop an interpretable model to identify biomarkers and predict subarachnoid hemorrhage based on a decision tree and feature selection method. However, my concerns are as follows:

  1. In this manuscript, samples are classified into three categories: healthy controls, subarachnoid hemorrhage patients, and subarachnoid hemorrhage patients with vasospasm. However, the authors employed decision trees to construct three models for performing binary classification. Indeed, the decision tree algorithm has been demonstrated to be a valuable tool in the context of multi-classification research. It is therefore recommended that authors employ decision trees directly in order to construct models for the classification of three types of samples.
  2. In order to promote research in this field, authors should make their datasets publicly available. To facilitate comprehension, authors are advised to furnish comprehensive descriptions of their datasets within the main text, including the number of samples in each category.
  3. The Tables are not numbered in the order in which they appear.
  4. There are two Table 2. The first is located on the page 2, and the second is located on the page 3.
  5. Precision_macro, recall_macro, and F1-macro are commonly used to evaluate the performance of multi-classification models. However, the current study constructed three binary classification model.
  6. In addition to accuracy, precision,_macro recall_macro, and F1-score_macro, ROC_macro and PRC_macro as well as areas under the two curves are also used to evaluate the performance of the model.

Author Response

Comments 1: In this manuscript, samples are classified into three categories: healthy controls, subarachnoid hemorrhage patients, and subarachnoid hemorrhage patients with vasospasm. However, the authors employed decision trees to construct three models for performing binary classification. Indeed, the decision tree algorithm has been demonstrated to be a valuable tool in the context of multi-classification research. It is therefore recommended that authors employ decision trees directly in order to construct models for the classification of three types of samples.

Response 1: Thank you for pointing this out. We agree that decision trees are capable of handling multi-class classification problems. However, we deliberately applied a binary classification framework for the following reasons [https://www.ncbi.nlm.nih.gov/pmc/articles/PMC12052818/]:

  • The pairwise clinical questions we addressed (e.g., Control vs SAH, SAH vs CVS) are distinct and independently meaningful from a diagnostic point of view.
  • The limited sample size per group (n=22) would compromise the stability and generalizability of a multiclass model.
  • Binary decision trees allow for simpler, more interpretable rules, which is particularly advantageous in clinical biomarker settings.

This rationale has been added to the revised manuscript in the Materials and Methods section:

"Although decision tree classifiers support multi-class problems, we intentionally applied binary classification models to answer specific clinical questions (e.g., early SAH detection, CVS risk stratification) while improving model robustness given the small group sizes. Binary classification also yielded simpler and more interpretable decision rules [https://www.ncbi.nlm.nih.gov/pmc/articles/PMC12052818/] ."

Comments 2: In order to promote research in this field, authors should make their datasets publicly available. To facilitate comprehension, authors are advised to furnish comprehensive descriptions of their datasets within the main text, including the number of samples in each category.

Response 2: We agree with this important point. The exact number of samples in each group (n=22 for Control, SAH, and CVS) was already clearly stated in the original manuscript in the Materials and Methods section. Additionally, we have now clarified the data availability at the end of the manuscript by adding a Data Availability Statement indicating that the dataset will be made available upon reasonable request due to ethical constraints and patient confidentiality.

Comments 3: The Tables are not numbered in the order in which they appear. There are two Table 2. The first is located on page 2, and the second is located on page 3.

Response 3: Thank you for catching this inconsistency. We have revised the numbering of all tables in the manuscript to ensure sequential consistency.

Comments 4: Precision_macro, recall_macro, and F1-macro are commonly used to evaluate the performance of multi-classification models. However, the current study constructed three binary classification models.

Response 4: Thank you for your observation. While these macro-averaged metrics are indeed widely used in multi-class problems, they also offer advantages in binary classification, especially in clinical contexts where class imbalance may vary. Macro-averaging ensures equal class weighting, regardless of class size. This rationale is already explained in the manuscript:

"Macro-averaging was used to ensure equal weight was given to each class, regardless of sample size, which is particularly relevant in datasets with modest but balanced group sizes, as in this study. "

Comments 5: In addition to accuracy, precision_macro, recall_macro, and F1-score_macro, ROC_macro and PRC_macro as well as areas under the two curves are also used to evaluate the performance of the model.

Response 5: Thank you for the suggestion. While our primary evaluation relied on standard classification metrics, we acknowledge the value of ROC and PRC analyses. However, considering our focus on interpretability and clinical relevance, and the balance between clarity and metric redundancy, we decided not to include additional ROC_macro or PRC_macro values. We believe the current evaluation—using macro-averaged accuracy, precision, recall, and F1—already provides a sufficiently robust and informative assessment of model performance in the context of this study.

Reviewer 2 Report

Comments and Suggestions for Authors

The manuscript "Transparent Machine Learning Reveals Diagnostic Glycan Biomarkers in Subarachnoid Hemorrhage and Vasospasm" is a statistical sequel of Author's previous publication: J. Clin. Med. 202514(2), 465; https://doi.org/10.3390/jcm14020465

the upsurge of AI and ML technologies in present day research is evident. The authors tried to approach Glycan-based research related to Subarachnoid Hemorrhage and Vasospasm with ML. This is a glycomic approach for N-glycans.

I have some concerns for the present manuscript:

From the reader's perspective the authors must elaborate the background of the samples/patient cohort and earlier findings. It is not expected that the reader will always be willing to go back and check the biochemistry to their original paper.

Please clearly state cohort size, and glycan structures clearly (not only the abbreviated).

Authors must realize that glycan structures they are dealing with came from MS and MS/MS data matched with a database. Accuracy of this kind of annotations are faulty since, mass spec does not characterize stereochemistry and carbohydrates differ largely by stereochemistry: for example, if database annotates fucose, it can easily be rhamnose.

Although the authors showed the statistical operations, their outcomes as specific biomarkers (quantitative/qualitative) need to be verified with unknown clinical cohorts.

As a ML based approach to metabolomic workflow, this manuscript should undergo some revision. As far as glycans are concerned, it may need much deeper biochemical/structural endeavors in future.

Author Response

Thank you for your valuable feedback regarding our manuscript on the diagnostic potential of serum N-glycans in subarachnoid hemorrhage (SAH) and cerebral vasospasm (CVS). We appreciate your insights and would like to address the specific concerns raised.

Comment 1:From the reader's perspective the authors must elaborate the background of the samples/patient cohort and earlier findings. It is not expected that the reader will always be willing to go back and check the biochemistry to their original paper.

Reply 1:

Our current research builds upon previous findings published in our earlier paper (Czabajszki et al., 2025), where we established the foundational understanding of glycosylation changes in SAH and CVS. In that study, we demonstrated that specific N-glycan profiles are altered in these patients compared to healthy controls, highlighting the potential of glycosylation as a biomarker for early diagnosis. We will reference these earlier findings more explicitly in our manuscript to connect the dots for readers and provide a cohesive narrative that underscores the significance of our current work.

Comment 2: Please clearly state cohort size, and glycan structures clearly (not only the abbreviated).

Reply 2: We acknowledge the importance of clearly stating the cohort size. In our study, we analyzed serum samples from 22 healthy controls, 22 SAH patients, and 22 patients with confirmed vasospasm. We will elaborate on the demographic and clinical characteristics of these cohorts in the revised manuscript to provide a clearer context for our findings.

Comment 3: Authors must realize that glycan structures they are dealing with came from MS and MS/MS data matched with a database. Accuracy of this kind of annotations are faulty since, mass spec does not characterize stereochemistry and carbohydrates differ largely by stereochemistry: for example, if database annotates fucose, it can easily be rhamnose.

Reply 3: Rhamnose is not typically found in N-glycans. N-glycans generally consist of a core structure that includes mannose and N-acetylglucosamine, and they can have various branching structures with additional sugars such as galactose, fucose, and sialic acid. While rhamnose is a sugar that can be found in other glycan types (such as O-glycans), it is not a standard component of N-glycan structures.

Comment 4: Although the authors showed the statistical operations, their outcomes as specific biomarkers (quantitative/qualitative) need to be verified with unknown clinical cohorts.

Reply 4: Your comment regarding the need for verification of our biomarkers in independent clinical cohorts is crucial. We agree that further validation is necessary to establish the clinical utility of the identified glycan markers. We will expand on the implications of our findings and the steps needed for future validation studies.

Comment 5: As a ML based approach to metabolomic workflow, this manuscript should undergo some revision. As far as glycans are concerned, it may need much deeper biochemical/structural endeavors in future.

Reply 5: We appreciate your suggestion for deeper biochemical and structural analysis in the context of our machine learning approach. We will include a discussion on potential biochemical pathways that may influence glycan alterations in SAH and CVS, and how these may be integrated into machine learning models for enhanced diagnostic precision.

In summary, we appreciate your constructive feedback and will incorporate these suggestions to improve the clarity and robustness of our manuscript. Thank you for helping us enhance the quality of our research.

Round 2

Reviewer 1 Report

Comments and Suggestions for Authors

This manuscript has been revised.